# DyRo-MCTS: A Robust Monte Carlo Tree Search Approach to Dynamic Job Shop Scheduling

## Abstract

Dynamic job shop scheduling, a fundamental combinatorial optimisation problem in various industrial sectors, poses substantial challenges for effective scheduling due to frequent disruptions caused by the arrival of new jobs. State-of-the-art machine learning methods have been used to learn scheduling policies that can make prompt, robust decisions in response to dynamic disturbances. However, these offline-learned policies are often imperfect, necessitating the use of planning techniques such as Monte Carlo Tree Search (MCTS) to improve performance at online decision time. The unpredictability of new job arrivals complicates online planning, as decisions based on incomplete problem information are vulnerable to disturbances. To address this issue, we propose the Dynamic Robust MCTS (DyRo-MCTS) approach, which integrates action robustness estimation into MCTS. DyRo-MCTS guides the production environment toward states that not only yield good scheduling outcomes but are also easily adaptable to future job arrivals. Extensive experiments show that DyRo-MCTS significantly improves the performance of offline-learned robust scheduling policies with acceptable online planning time. Moreover, DyRo-MCTS consistently outperforms state-of-the-art MCTS algorithms across various dynamic scheduling scenarios. Further analysis reveals that its ability to make robust scheduling decisions leads to long-term, sustainable performance gains under disturbances.

## 1 Introduction

Dynamic job shop scheduling (DJSS) is an NP-hard combinatorial optimisation problem that is prevalent across various industry sectors (Wang et al., 2020b). It involves determining the processing order of jobs on machines to minimise objectives such as job tardiness. Unlike static scheduling tasks, which assume all jobs are available on the shop floor from the outset, dynamic scheduling accounts for the real-time arrival of new jobs. This feature of DJSS introduces two major challenges in scheduling. First, complete information for planning is unavailable, as the details of the new jobs are only revealed upon their release. Second, it demands rapid responses to dynamic events to prevent production slowdowns while awaiting scheduling decisions.

Consequently, exact optimisation methods (Brucker et al., 1994) and meta-heuristic approaches (Davis, 1985) are less suitable for dynamic scheduling due to their considerable time consumption for exhaustively optimising the solution (Mohan et al., 2019). Moreover, solutions optimised based on the current job information would not necessarily remain optimal if information about future job arrivals could be taken into account during planning.

To promptly respond to scheduling needs in dynamic production environments, state-of-the-art approaches aim to design real-time scheduling systems operating in a completely reactive manner (Renke et al., 2021). They make immediate job selection decisions when a machine becomes idle, based on the estimated priorities $\boldsymbol{\pi} = \{\pi_1, ..., \pi_n\}$ of the $n$ candidate jobs in the machine buffer.

In existing studies (Xu et al., 2025), $\boldsymbol{\pi}$ is estimated by learning scheduling policies through machine learning (ML) techniques such as deep reinforcement learning (DRL) and genetic programming (GP). These methods approximate a function $f(s) \rightarrow \boldsymbol{\pi}$ that takes features of the state $s$ as input,

aiming to make decisions that are not only fast but also robust. By simulating dynamic events during training, the ML models recognise the patterns of the dynamic environment and thus make robust decisions that tolerate disturbances. However, the learned policies are often imperfect in practice, due to the inherent challenges of the training and the limited representational capacity of the extracted features. Moreover, generalising the policy to the vast range of unseen states in the job shop environment remains a significant challenge.

Given that the raw output of an offline policy is often suboptimal, we use it as a prior to guide a Monte Carlo Tree Search (MCTS) for improving the estimation of job priorities $\pi$ at online decision time. MCTS performs selective lookahead searches from each encountered state and estimates $\pi$ based on the posterior statistics collected during the search.

However, in DJSS, we lack complete information of the problem for planning because the details of new jobs are unforeseen. Explicitly modelling this transition uncertainty using existing methods (Kohankhaki et al., 2024) is challenging due to the infinite possibilities of new job arrivals. Sample-based approaches (Silver & Veness, 2010) are also impractical due to the NP-hard nature of DJSS. Each randomly introduced job factorially expands the search space, hindering real-time decision-making. Therefore, existing MCTS-based dynamic scheduling methods (Li et al., 2021; He et al., 2025; Kim & Kim, 2022) often compromise by restricting their online planning to existing jobs. However, our empirical studies show that even when guided by priors learned from robust scheduling methods, an MCTS algorithm that ignores transition uncertainty in DJSS still tends to greedily optimise the schedule into a state that is difficult to tolerate disturbances.

To address this issue, we propose the Dynamic Robust MCTS (DyRo-MCTS) algorithm, which not only plans for good outcomes (minimum job tardiness) based on current environment information, but also guides production towards states that remain easily adjustable when new jobs arrive. In addition to estimating the action value $q(s, a)$, DyRo-MCTS also estimates the action robustness $\rho(s, a)$ with respect to unforeseen disturbances. Inspired by the work of Branke & Mattfeld (2005), we calculate $\rho(s, a)$ based on the distribution of machine utilisation. A Dynamic Robust Upper Confidence bound for Trees (DyRo-UCT) is introduced, demonstrating strong empirical performance in balancing exploration, exploitation, and robustness in online search.

The overall goal of this research is to develop a robust online planning algorithm for DJSS. Our contributions are as follows.

1. We propose DyRo-MCTS, a lookahead search algorithm that enable state-of-the-art dynamic scheduler with online planning ability, allowing non-myopic decision-making.

2. To enhance existing MCTS-based algorithms for robust planning in DJSS, we integrate an easy-to-implement yet effective robustness estimation mechanism into the tree policy of MCTS, achieving strong empirical performance.

3. Empirically, we show that the performance of offline-learned robust scheduling policies can be substantially improved with acceptable online planning time. We also highlight the importance of making robust scheduling decision under disturbances: maintaining a production environment that is easily adaptable to new jobs leads to long-term benefits.

## 2 BACKGROUND

### 2.1 DYNAMIC JOB SHOP SCHEDULING

DJSS is an NP-hard combinatorial optimisation problem characterised by the continual arrival of new jobs over time. Each job $J_i$ has a due date $d_i$ and a user-defined weight $w_i$, and consists of a sequence of operations $\{O_{i1}, O_{i2}, ...\}$ that must be performed in a predefined order. Each operation $O_{ij}$ must be processed on a specific machine for a fixed duration $p_{ij}$ without interruption, and each machine can handle only one operation at a time. New job arrivals are random events that follow a Poisson distribution. The details of a new job—such as its arrival time, due date, and the processing time of its operations—are only revealed upon release. The scheduling objective is to minimise the mean weighted tardiness $\mathcal{T}$ of jobs over a long scheduling horizon:

$$\mathcal{T} = \frac{1}{|\mathcal{J}|} \sum_{i \in \mathcal{J}} \left( w_i \cdot \max\{c_i - d_i, 0\} \right),$$

where $\mathcal{J}$ is the set of all jobs arriving during the scheduling horizon, and $c_i$ is the completion time of $J_i$.

## 2.2 SCHEDULE ROBUSTNESS IN DJSS

Robust scheduling aims to optimise the expected quality of a schedule in an uncertain environment. Anticipating dynamic changes and keeping the schedule flexible is an effective way to enhance schedule robustness. Branke & Mattfeld (2005) suggested that schedule flexibility is related to the distribution of machine utilisation.

Consider the two schedules in Figure 1, where two jobs with due dates at time 9 are to be scheduled. Schedules (a) and (b) are equivalent in terms of total job tardiness, with each resulting 1 unit. However, they differ in robustness under dynamic disturbances. A new job may arrive at any moment, and the production system will follow the existing schedule until that point. Consequently, the portion of the schedule near the current time is more likely to be executed as planned, whereas the later part of the schedule is more likely to be changed to incorporate new jobs. In this respect, schedule (b) is

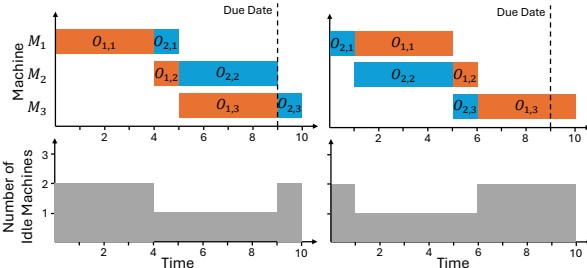

(a) Less Robust Schedule     (b) More Robust Schedule

Figure 1: Comparison of schedule robustness. The Gantt charts are shown above, with the corresponding distributions of machine idleness over time shown below.

more easily adjustable, as it allocate more intensive machine utilisation in the early stage, completing more operations before the need of rescheduling arises. In contrast, schedule (a) leaves more machines idle early on, increasing the chance that these resources are wasted before they can be allocated to new jobs. Moreover, it postpones more operations to be processed later. These operations may become a backlog by the time rescheduling is required. Thus, a simple way to maintain schedule robustness is to avoid early machine idleness.

## 2.3 MONTE CARLO TREE SEARCH

MCTS is a heuristic search algorithm widely used in decision-making problems (Świechowski et al., 2023). While MCTS can operate without prior knowledge, it often leverages human-designed or ML-based policies to guide search (Kemmerling et al., 2024b). In MCTS, nodes represent states, and edges represent actions. The algorithm iteratively builds a search tree through following four key steps.

(1) SELECTION. Starting from the root node, a tree policy guides the traversal to a leaf node for expansion. A widely used tree policy is the Predictor Upper Confidence bound for Trees (PUCT), which extends the UCT strategy by incorporating prior probabilities (Silver et al., 2017). PUCT selects a child node based on the formula:

$$a^* = \arg\max_a \left[ q(s,a) + c \cdot p(s,a) \frac{\sqrt{n(s)}}{1 + n(s,a)} \right],$$

where $q(s,a)$ is the value of choosing action $a$ at state $s$. $p(s,a)$ is the prior probability indicating promising search directions. $n(s)$ is the visit count of state $s$, and $n(s,a)$ denotes the number of times action $a$ has been selected. The left term of the sum promotes exploitation of high-value actions, while the right term encourages exploration of less-visited actions. A tunable constant $c$ controls the trade-off between exploitation and exploration.

(2) EXPANSION. When a leaf node that is not fully expanded is reached during traversal, the expansion step adds one or more child nodes representing previously unvisited states. The expanded nodes are selected either randomly or guided by a policy.

(3) EVALUATION. The value of each newly expanded node is estimated via Monte Carlo rollouts. One or more trajectories are simulated from the node until a terminal state is reached, yielding a

reward $r$. If a reliable value function is accessible, it can also be employed to evaluate the newly expanded nodes.

(4) BACKPROPAGATION. The evaluation result is propagated backward through the traversed nodes, incrementing the visit count $n(s, a) \leftarrow n(s, a) + 1$ and updating the total action value $w(s, a) \leftarrow w(s, a) + r$ along the path. The mean action value is then computed as $q(s, a) \leftarrow \frac{w(s,a)}{n(s,a)}$.

After completing MCTS, the next move can be determined by selecting either the most visited action from the root state or the action with the highest $q(s, a)$ value. Compared with offline-learned policies, MCTS focuses on each online encountered state and does not pursue generalisation. Its lookahead on future states enables it to make strong and informed moves on-the-fly.

## 2.4 RELATED WORK

**Offline Learning of Robust Scheduling Policies.** State-of-the-art methods for DJSS employ ML techniques to automatically learn robust scheduling policies (Renke et al., 2021), often outperforming heuristics manually designed by human experts (Holthaus & Rajendran, 2000). Among these, DRL and GP are two widely adopted approaches (Xu et al., 2025).

DRL employs artificial neural networks to approximate policy and value functions (Sutton & Barto, 2018). To learn scheduling policies, different DRL algorithms can be used, such as DQN (Liu et al., 2023) and PPO (Zhang et al., 2020; Song et al., 2023). DRL-based policies can operate by selecting low-level heuristics (Liu et al., 2022) or achieve end-to-end policy learning through integration with graph neural networks (Liu et al., 2024).

GP, on the other hand, is an evolutionary computation approach to learning scheduling policies (Zhang et al., 2021). GP-based policies can adopt various representations, with tree-based (Chen et al., 2025) and linear structures (Huang et al., 2023) being the most commonly used. These representations offer flexibility in shape and depth, allowing the search space to accommodate a wide range of high-quality scheduling policies. GP is also well known for its ability to learn interpretable policies, which supports users in making more reliable and transparent scheduling decisions (Pang et al., 2024).

**Online Planning in Scheduling via MCTS.** MCTS is widely used in scheduling and combinatorial optimisation. In static scheduling, it is typically used to incrementally construct solutions. Wang et al. (2020a) employed MCTS to solve the parallel machine scheduling problem, where a neural network policy was trained through PPO and then used to guide MCTS. This method can achieve better performance compared with meta-heuristics. Saqlain et al. (2023) applied MCTS to flexible job shop scheduling, demonstrating that MCTS offers greater advantages in complex scenarios involving a larger number of jobs. Kemmerling et al. (2024b) provided a comprehensive review of neural MCTS applications beyond games and conducted an in-depth investigation into its use for static job shop scheduling (Kemmerling et al., 2024a). Through extensive experiments, they tested different MCTS component designs for static job shop scheduling.

For solving dynamic scheduling problems, existing methods primarily focus on designing effective response strategies, adopting well-established MCTS techniques to improve search efficiency and decision speed. Li et al. (2021) adopted subtree keeping and RAVE when designing the MCTS planner for DJSS, with prior knowledge initialised and updated on the fly and rolling time windows used to accelerate decisions. Additional MCTS search techniques, including multi-branching simulation and subtree pruning, have also been explored for solving DJSS (He et al., 2025). Another major direction of research is adapting MCTS to different DJSS variants, including problems with flexible machines, transportation-time constraints and automated guided vehicles (Kim & Kim, 2022), as well as other related task-scheduling problems (Cheng et al., 2019; Li et al., 2022).

In existing studies, although numerous MCTS techniques have been applied to DJSS for efficient online search, the long-term scheduling performance can still deteriorate as dynamic disturbances accumulate. Lookahead planning in DJSS can only relies on incomplete information, as future job arrivals are unforeseen and difficult to model with uncertainty-aware MCTS methods (Kohankhaki et al., 2024). To address this limitation, we develop an easy-to-implement yet highly effective mechanism that enables MCTS to make robust decisions for dynamic scheduling scenarios.

## 3 PROPOSED METHOD

This section describes the proposed DyRo-MCTS algorithm, designed to perform robust online planning for DJSS using imperfect problem information. We begin by formulating the DJSS problem as a Markov decision process (MDP), then describe how DyRo-MCTS makes online decisions. Lastly, we introduce the action robustness estimation mechanism of DyRo-MCTS, which enables more reliable decision-making under job arrival disturbances.

### 3.1 MARKOV DECISION PROCESS FORMULATION

Unlike the MDP formulation in static scheduling, where a complete schedule is constructed incrementally with the partial schedule at each step, the DJSS problem is typically formulated as a MDP driven by discrete-event simulation (Turgut & Bozdag, 2020).

A **state** corresponds to a moment in the job shop when at least one machine becomes idle and multiple candidate jobs are present in its buffer. The features extracted to represent the state depend on the design of the specific ML method, with manually crafted features (Huang et al., 2024) and graph-based features (Liu & Huang, 2023) being the most common.

An **action** involves selecting a candidate job for processing on the idle machine. The action space varies with the number of candidate jobs in the buffer. This ensures that the precedence and resource constraints of DJSS are always satisfied by the generated schedule.

The **state transition** in DJSS is stochastic. After executing an action, the system may transition to infinitely many possible next states due to random future job arrivals. Since this transition uncertainty is difficult to model explicitly, and randomly sampling new jobs would dramatically expand the search space, we restrict the lookahead search of MCTS to the set of existing jobs on the shop floor. This allows planning to proceed under a deterministic transition model and prevents the search tree from expanding without bound in breadth and depth. The proposed DyRo-MCTS algorithm is designed to compensate for the estimation bias introduced by this transition model approximation.

At the terminal state, the **reward** $r$ is defined as $r = -\mathcal{T}$, reflecting the objective of minimising the mean weighted tardiness $\mathcal{T}$.

### 3.2 DYRO-MCTS ALGORITHM

The framework of DyRo-MCTS is illustrated in Figure 2. At each online decision point $\dot{s}_t$, an MCTS is executed. To ensure timely responses in DJSS, the goal of MCTS in our approach is not to exhaustively optimise a schedule, but rather to produce real-time job priority estimates $\boldsymbol{\pi}$ for determining the immediate action $a_t$. $\boldsymbol{\pi}$ is considered sufficiently strong after $N_{mcts}$ iterations of MCTS. In practice, the actual number of executed iterations may be fewer than $N_{mcts}$, as some nodes from the previous decision step $\dot{s}_{t-1}$ can be reused at $\dot{s}_t$.

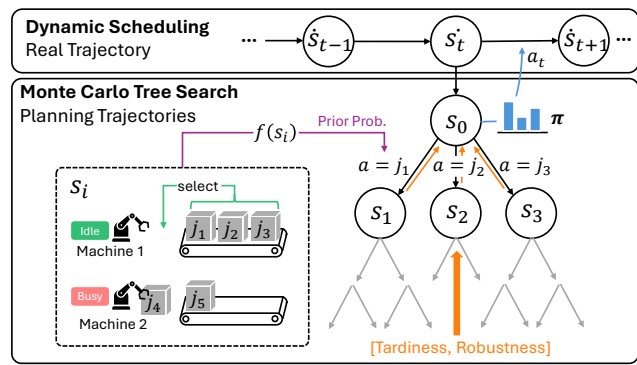

Figure 2: The DyRo-MCTS framework for DJSS.

When no new job arrives between two consecutive decision points, the entire search tree from the subsequent state is retained, and all statistics, including total action value $w(s, a)$, visit count $n(s)$, prior $p(s, a)$, and normalisation ranges, are preserved. Once a new job arrives, the whole search tree is discarded, as the previous rollouts do not contain the operations of the new job, so their statistics are not reusable.

This work adopts two ML methods for learning policies offline. One is a DRL-based method proposed in Liu et al. (2023); the other is a GP-based method in the work of Chen et al. (2025). The learned policy provides prior probabilities $p(s, a)$ in PUCT, prioritises node expansion, and guides rollout simulations. In theory, any prior probability—whether derived from domain knowledge or

learned through ML—can be used. In this paper, we refer to the vanilla prior-guided MCTS method as PUCT-MCTS.

In DJSS, the problem information is incomplete at decision points. Greedily selecting the action with the highest $q(s, a)$ may appear beneficial at the current step but can gradually steer the production system towards states that are difficult to schedule new jobs. In this regard, a good action should not only have a high value $q(s, a)$ but also exhibit sufficient robustness $\rho(s, a)$ to tolerate disturbances. The key mechanism by which DyRo-MCTS balances action value and robustness is through a DyRo-UCT strategy

$$a^* = \arg\max_a \left[ \mathcal{E}(s, a) + c \cdot p(s, a) \frac{\sqrt{n(s)}}{1 + n(s, a)} \right]$$

$$\mathcal{E}(s, a) = \alpha \cdot q(s, a) + (1 - \alpha) \cdot \rho(s, a)$$

which is derived from PUCT with a single modification: the exploitation term $\mathcal{E}(s, a)$ is adjusted to interpolate between $q(s, a)$ and $\rho(s, a)$, controlled by a parameter $\alpha \in [0, 1]$.

The proposed DyRo-UCT influences the entire search process. Directly, it allocates more search resources to regions with both high value and robustness. Ultimately, it influences the final move decision by affecting the action visit count $n(s_0, a)$ from root node $s_0$. We estimate $\boldsymbol{\pi}$ as $\pi_a \propto n(s_0, a)$, and the action with highest $\pi_a$ is selected.

### 3.3 ACTION VALUE AND ROBUSTNESS

Action value $q(s, a)$ reflects the ultimate goal of scheduling: minimising tardiness. Since the range of tardiness in DJSS is unbounded, and $q(s, a)$ and $\rho(s, a)$ must be constrained to [0, 1] to ensure the effectiveness of the DyRo-UCT strategy, we scale the tardiness $\mathcal{T}_i$ of a schedule with the maximum tardiness $\mathcal{T}_{\max}$ and minimum tardiness $\mathcal{T}_{\min}$ recorded during the tree search. $q(s, a)$ is estimated by averaging the tardiness of schedules resulting from action $a$:

$$q(s, a) = \frac{1}{N} \sum_i^N \frac{\mathcal{T}_{\max} - \mathcal{T}_i}{\mathcal{T}_{\max} - \mathcal{T}_{\min}}.$$

Action robustness $\rho(s, a)$ reflects the tolerance to disturbances caused by new job arrivals. As introduced in Section 2.2, the robustness of a schedule is related to the distribution of machine utilisation, with lower machine idleness in the early stages being more desirable. Accordingly, we formulate the robustness $\mathcal{R}$ of a schedule as the integral of weighted machine idleness across the entire scheduling period:

$$\mathcal{R} = \sum_{m \in \mathcal{M}} \int_0^T w(t) \cdot \mathbb{I}_m(t) \, dt,$$

where $m$ is a machine in the machine set $\mathcal{M}$. $t$ denotes the time in the schedule, with $T$ representing the makespan. $\mathbb{I}_m(t)$ is an indicator function that returns 1 if machine $m$ is idle at time $t$, and 0 otherwise. $w(t)$ is a weighting function for penalising early idleness—i.e., it assigns smaller negative values to idleness occurring at smaller $t$. In this study, $w(t)$ is implemented as a modified form of the rectified linear function (Branke & Mattfeld, 2005):

$$w(t) = \min\left(0, \frac{t}{\beta} - 1\right).$$

The above $w(t)$ function is controlled by a single parameter $\beta$, making it easy to tune. Moreover, it confines its effect to the time range $[0, \beta]$, preventing the algorithm from searching schedules with excessive idleness beyond $\beta$ in pursuit of a large $\mathcal{R}$.

Finally, the robustness $\rho(s, a)$ of performing action $a$ at state $s$ is estimated through Monte Carlo method:

$$\rho(s, a) = \frac{1}{N} \sum_i^N \frac{\mathcal{R}_i - \mathcal{R}_{\min}}{\mathcal{R}_{\max} - \mathcal{R}_{\min}},$$

where $\mathcal{R}_i$ is the robustness of the $i^{th}$ rollout from action $a$. $\mathcal{R}_{\max}$ and $\mathcal{R}_{\min}$ are the maximum and minimum values of $\mathcal{R}$ recorded during the tree search.

This robustness estimation is easy to implement, as it does not involve any additional learning process and only requires the additional step of recording machine idle time. Its computational overhead compared with PUCT-MCTS is negligible, yet it leads to a significant performance improvement, as demonstrated in the experiments introduced in the next section.

## 4 EXPERIMENTAL STUDIES

### 4.1 EXPERIMENT DESIGN

This paper adopts a widely recognised DJSS simulation configuration (Zhang et al., 2024) to evaluate the proposed algorithm. In each simulation, new jobs continuously arrive at the job shop and are processed by 10 different machines. The number of operations per job follows a discrete uniform distribution $U(2, 10)$, and the processing time of each operation is drawn from $U(1, 99)$. New jobs arrive according to a Poisson process with a average rate $\lambda$. A widely adopted approach to control the arrival frequency is to use a utilisation parameter $u$, where a higher value indicates a busier job shop. The value of $\lambda$ can be calculated as $(|\mathcal{M}| \cdot u)/\bar{p}$, where $\bar{p}$ is the mean processing time of jobs, and $|\mathcal{M}|$ is the number of machines. The first 1000 arriving jobs are used to warm up the job shop environment, ensuring that testing is conducted under stable operating conditions. The tardiness of the subsequent 5000 jobs is recorded to evaluate the performance of the algorithm. We consider two scheduling objectives. The first, denoted $T_{mean}$, assumes equal job weights ($w_i = 1$ when calculating $\tau$). The second, $WT_{mean}$, assigns different weights to jobs: 20%, 60%, and 20% of the jobs have weights of 1, 2, and 4, respectively. Unless otherwise specified, the decision budget of MCTS (i.e. the number of search iterations $N_{mcts}$) is set to 100.

### 4.2 PARAMETER ANALYSIS

The DyRo-MCTS algorithm introduces two key parameters, $\alpha$ and $\beta$, to control a robust lookahead search in MCTS.

The parameter $\alpha$ serves as the interpolation parameter in the DyRo-UCT strategy, balancing the action value and robustness. A higher $\alpha$ emphasizes the action value, reducing the influence of action robustness. Setting $\alpha = 1$ removes all modifications introduced by DyRo-MCTS, reverting to the PUCT-MCTS algorithm.

The parameter $\beta$ governs the slope of the linear weighting function for machine idleness. A higher $\beta$ distributes the weight of machine idleness more evenly across the entire schedule, while a lower $\beta$ penalises early machine idleness more severely.

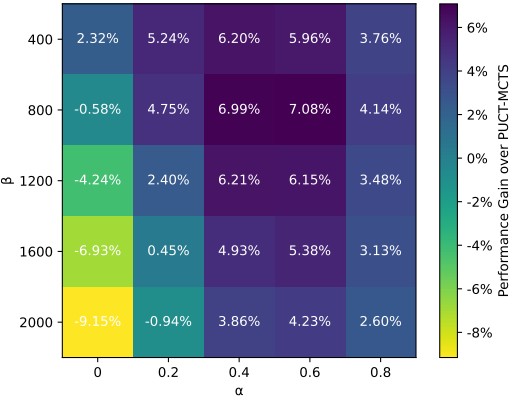

Figure 3: Performance gain (the higher the better) of DyRo-MCTS over PUCT-MCTS under different settings of $\alpha$ and $\beta$.

Parameter tuning is performed on a separate validation set comprising 30 distinct scheduling instances, disjoint from the test set. The results are presented in Figure 3, with the values in the heatmap indicating the performance improvement of DyRo-MCTS over the vanilla prior-guided MCTS (i.e., $\alpha = 1$).

The results show that DyRo-MCTS consistently yields performance improvements across a wide range of parameter settings, as indicated by the widespread presence of positive values in the heatmap. This demonstrates that the method is tolerant to parameter variation and can be safely applied without requiring precise tuning.

The performance of DyRo-MCTS is more sensitive to the parameter $\alpha$. The best performance is observed when $\alpha$ is in the range of 0.4 to 0.6, where the influences of action value and robustness are nearly equally considered. Performance deterioration (i.e., negative performance gain) occurs only in the lower-left corner of the heatmap, where the algorithm strongly favours actions with high robustness value (i.e., $\alpha$ close to 0), yet the robustness estimates across actions are not sufficiently

Table 1: Performance comparison of pure offline policy, PUCT-MCTS, and DyRo-MCTS guided by Random, Manual, DRL, and GP policies across different scheduling scenarios.

| Scenario | Policy | w/o Online Planning | + PUCT-MCTS | | + DyRo-MCTS | |
|---|---|---|---|---|---|---|
| | | Perf. | Perf. | Imp. | Perf. | Imp. |
| $T_{mean}$ 0.85 | Random | $919.51 \pm 3.69$ | $499.65 \pm 1.45 (\uparrow)$ | 46% | $434.46 \pm 1.15 (\uparrow)(\uparrow)$ | 53% |
| | Manual | $673.61 \pm 145.32$ | $484.83 \pm 36.47 (\uparrow)$ | 26% | $434.36 \pm 28.77 (\uparrow)(\uparrow)$ | 34% |
| | DRL | $608.94 \pm 72.22$ | $460.39 \pm 14.79 (\uparrow)$ | 24% | $425.78 \pm 15.73 (\uparrow)(\uparrow)$ | 29% |
| | GP | $442.31 \pm 2.14$ | $404.24 \pm 5.54 (\uparrow)$ | 9% | $391.9 \pm 11.35 (\uparrow)(\uparrow)$ | 11% |
| $T_{mean}$ 0.95 | Random | $3241.55 \pm 12.48$ | $2388.03 \pm 8.80 (\uparrow)$ | 27% | $1872.57 \pm 4.45 (\uparrow)(\uparrow)$ | 43% |
| | Manual | $2268.15 \pm 584.36$ | $1734.03 \pm 165.68 (\uparrow)$ | 21% | $1435.60 \pm 120.24 (\uparrow)(\uparrow)$ | 34% |
| | DRL | $2011.89 \pm 254.28$ | $1554.71 \pm 50.24 (\uparrow)$ | 22% | $1345.65 \pm 31.42 (\uparrow)(\uparrow)$ | 32% |
| | GP | $1380.21 \pm 13.42$ | $1299.95 \pm 15.41 (\uparrow)$ | 6% | $1218.58 \pm 21.53 (\uparrow)(\uparrow)$ | 12% |
| $WT_{mean}$ 0.85 | Random | $2023.41 \pm 8.06$ | $1013.17 \pm 2.39 (\uparrow)$ | 50% | $878.91 \pm 2.13 (\uparrow)(\uparrow)$ | 57% |
| | Manual | $1381.52 \pm 388.72$ | $956.67 \pm 114.25 (\uparrow)$ | 28% | $862.71 \pm 84.53 (\uparrow)(\uparrow)$ | 34% |
| | DRL | $1091.84 \pm 112.19$ | $824.33 \pm 27.04 (\uparrow)$ | 24% | $783.96 \pm 28.5 (\uparrow)(\uparrow)$ | 28% |
| | GP | $765.65 \pm 6.22$ | $711.05 \pm 14.72 (\uparrow)$ | 7% | $698.32 \pm 21.37 (\uparrow)(\uparrow)$ | 9% |
| $WT_{mean}$ 0.95 | Random | $7136.24 \pm 28.05$ | $5098.26 \pm 18.21 (\uparrow)$ | 30% | $3967.47 \pm 15.34 (\uparrow)(\uparrow)$ | 45% |
| | Manual | $4452.71 \pm 1676.25$ | $3234.54 \pm 515.63 (\uparrow)$ | 22% | $2713.14 \pm 363.27 (\uparrow)(\uparrow)$ | 34% |
| | DRL | $3216.09 \pm 251.21$ | $2677.95 \pm 112.63 (\uparrow)$ | 17% | $2391.19 \pm 68.58 (\uparrow)(\uparrow)$ | 26% |
| | GP | $2143.82 \pm 29.36$ | $2054.56 \pm 81.01 (\uparrow)$ | 4% | $1950.80 \pm 58.38 (\uparrow)(\uparrow)$ | 9% |

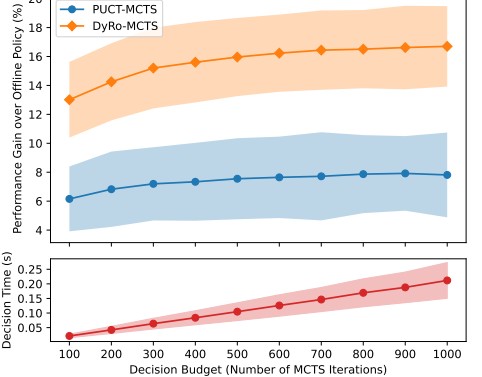

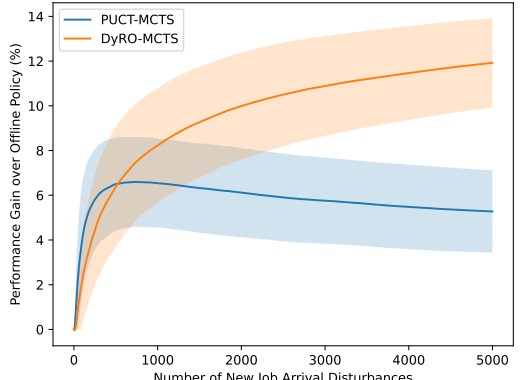

Figure 4: Scaling of performance gains of PUCT-MCTS and DyRo-MCTS with increasing decision budget.

Figure 5: Comparison of performance gains between DyRo-MCTS and PUCT-MCTS under continuously occurring job arrival disturbances.

distinct (i.e., high $\beta$), leading to less effective guidance. We ultimately select $\alpha = 0.6$, $\beta = 800$. The exploration constant $c$ is also tuned and set to 3 (see Appendix A for details).

### 4.3 MAIN RESULTS

In this section, we evaluate the performance of DyRo-MCTS across various dynamic scheduling scenarios (as shown in Table 1). These scenarios involve two scheduling objectives: minimising mean tardiness ($T_{mean}$) and mean weighted tardiness ($WT_{mean}$), under two utilisation levels (0.85 and 0.95). Since the goal is to minimise tardiness, lower values in the Perf. column indicate better performance.

A practical online planning algorithm should be compatible with offline policies obtained through different methods. In this paper, we consider four widely studied scheduling methods for DJSS. Random denotes a baseline with no prior knowledge, where jobs are scheduled randomly. Manual refers to the average results of ten scheduling heuristics designed by human experts (heuristic descriptions and detailed results for each heuristic are provided in the Appendix B). For the ML-based methods: GP (Chen et al., 2025) and DRL (Liu et al., 2023), we trained 30 policies using per method. Each policy is tested on 30 instances, resulting in 900 independent MCTS runs. We

| Algorithm | PSAO | Li-MCTS | DRL | GP | PUCT-MCTS | DyRo-MCTS |
|---|---|---|---|---|---|---|
| Decision Time (ms) | $8.50 \times 10^3$ | 68.01 | 0.50 | $7.97 \times 10^{-3}$ | 20.53 | 22.30 |

Table 2: Decision-time comparison among different robust scheduling algorithms.

perform the Wilcoxon signed-rank test to assess statistical significance; a significant improvement is marked as ($\uparrow$).

The results in Table 1 show that the PUCT-MCTS can already significant improve the performance of offline-learned policies. With the robust lookahead planning capability of DyRo-MCTS, the performance of these policies can be further enhanced, showing significant improvement over PUCT-MCTS.

We also observe that policies with poor standalone performance typically have greater potential for improvement via online planning. For instance, DyRo-MCTS with random rollouts can yield up to a 57% improvement over purely random scheduling. However, the quality of the guiding policy remains critical for achieving strong overall performance. A further analysis of the impact of offline policy quality on online planning is provided in Appendix C. In our experiments, GP produces the best policies, and DyRo-MCTS guided by these policies achieved the best results across all scenarios. Therefore, our further analysis of DyRo-MCTS adopts GP-based policies.

### 4.4 COMPARISON WITH SOTA DYNAMIC SCHEDULING METHODS

In this section, we compare DyRo-MCTS with five state-of-the-art dynamic scheduling methods (shown in Figure 6). PSAO (Duan & Wang, 2022) is a multi-objective online planner that combines particle swarm optimisation with arithmetic optimisation to improve schedule quality and robustness simultaneously. Li-MCTS is an MCTS algorithm for DJSS proposed by Li et al. (2021), incorporating several advanced online planning techniques. Two ML-based robust scheduling methods (RL and GP), together with PUCT-MCTS, are also included. All methods follow the configurations reported in their original papers.

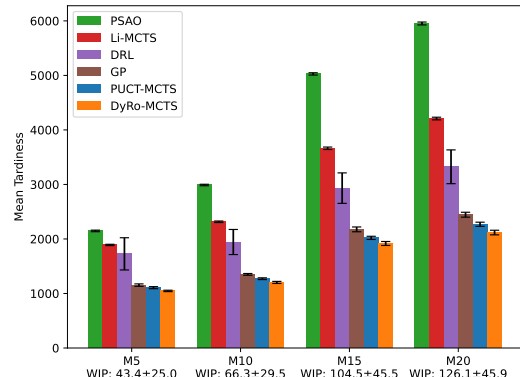

To examine the performance of an algorithm across different problem scales, we vary both the number of machines and the Work-in-Progress (WIP). For a fixed machine utilisation level of 0.95, increasing the number of machines leads to more concurrent jobs (i.e., higher WIP) and thus a more challenging scheduling environment. The x-axis of Figure 6 shows the number of machines and the mean WIP recorded at each decision point.

The results show that, for all algorithms, mean tardiness increases as the problem scale grows, reflecting the difficulty of scheduling a large number of jobs across many machines. The two ML-based methods, RL and GP, generally out-

Figure 6: Comparison of six dynamic scheduling methods across four problem scales.

perform the traditional online planners PSAO and Li-MCTS. The best overall performance is achieved by PUCT-MCTS and DyRo-MCTS, both guided by state-of-the-art ML-based policies, with DyRo-MCTS delivering consistently superior results across all four problem scales.

Table 2 reports the decision times of the compared methods. Among them, only the meta-heuristic method PSAO requires a considerably long time to make a decision. Compared with PUCT-MCTS, DyRo-MCTS increases the decision time by only 2 ms for robustness estimation. This level of time consumption is acceptable in many real-world scheduling scenarios.

### 4.5 IMPACT OF DECISION BUDGET

The previous experiments used 100 MCTS iterations as the decision budget. It is worthwhile to examine whether increasing the budget $N_{mcts}$ can further enhance the performance of DyRo-MCTS and how time consumption scales with $N_{mcts}$. The results are presented in Figure 4.

According to Figure 4, the performance of both MCTS and DyRo-MCTS gradually improves as the decision budget increases, with 1000 iterations yielding significantly better results than 100 iterations, as confirmed by the Wilcoxon signed-rank test.

There is no notable difference in time consumption between PUCT-MCTS and DyRo-MCTS, so we report the average decision time for both methods in Figure 4. The results show that the time consumption of scheduling decision increases linearly with the number of MCTS iterations. In our main experiments, the configuration of 100 MCTS iterations takes $0.021 \pm 0.006$ seconds per decision. When increased to 1000 iterations, the decision time rises to $0.212 \pm 0.061$ seconds. The above-mentioned time consumption of DyRo-MCTS is negligible for many real-world DJSS applications (Yang et al., 2025).

### 4.6 PERFORMANCE ANALYSIS UNDER ONGOING JOB ARRIVAL DISTURBANCES

In this section, we analyse how DyRo-MCTS achieves better performance than PUCT-MCTS under the ongoing disturbances in dynamic scheduling. The challenge of DJSS lies in making decisions that can withstand continual job arrival disturbances while maintaining low job tardiness over an extended scheduling period. To this end, we monitor how the performance of each algorithm evolves as disturbances occur continuously.

We apply the offline policy, PUCT-MCTS, and DyRo-MCTS to the same scheduling instance, which undergoes 5000 disturbances during scheduling. At the release of each disturbance, we record the total job tardiness at the moment. This experiment is repeated 1000 times across different instances, and the averaged results are presented in Figure 5. Initially, PUCT-MCTS exhibits rapid performance growth, achieving higher performance gains than DyRo-MCTS in the early phase (approximately the first 600 disturbances). However, as disturbances continue to accumulate, the performance of DyRo-MCTS gradually surpasses that of PUCT-MCTS, leading to a progressively larger performance gap.

The slower performance growth of DyRo-MCTS is due to its DyRo-UCT selection strategy allowing some jobs to be delayed in order to maintain a production environment that is more adaptable to future job arrivals. As a result, although its performance improvement is gradual, the growth is sustained over the long run.

In contrast, PUCT-MCTS only aims to minimise job tardiness at every decision point. Initially, this approach appears effective, as it results in less tardiness and leads to a rapid performance growth. However, this improvement is difficult to sustain as more disturbances occur. The jobs scheduled for later execution eventually become backlogged when rescheduling is required. Consequently, its performance advantage gradually diminishes as the number of disturbances increases.

## 5 CONCLUSIONS

This paper aims to design a robust online planning algorithm for the DJSS problem. This goal has been achieved through the proposed DyRo-MCTS algorithm. We develop a simple yet effective action robustness estimation process for MCTS that avoids sampling numerous unpredictable dynamic events, guiding the environment towards states that better adapt to new job arrivals. Empirical analysis demonstrates that DyRo-MCTS outperforms both the offline policy and the vanilla prior-guided MCTS across different scheduling scenarios. Moreover, the performance of DyRo-MCTS continues to improve as the decision budget increases. Further analysis reveals that robust scheduling decisions enable DyRo-MCTS to achieve sustainable performance growth under disturbances.

Currently, most research in the area of DJSS focuses on designing effective ML algorithms for learning scheduling policies offline. This work makes a pioneering exploration into online planning and demonstrates its promise. This work highlights that a good dynamic scheduling system relies on the combined influence of three key factors: high-quality scheduling policies learned offline, effective lookahead search during online planning, and robust decision-making under incomplete problem information.

The initiative of considering action robustness during online planning is also worth to be investigated in other dynamic combinatorial optimisation problems. Our future work will focus on developing improved methods for learning more suitable policies to guide MCTS.

REPRODUCIBILITY

This research will be a open-source project upon publication of the paper, released under a license permitting free use for research purposes. The following will be made publicly available:

- All source code
- All raw experimental data
- Jupyter notebook files for analyzing the raw data and generating the figures presented in this paper

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

## A ADDITIONAL PARAMETER TUNING

### A.1 EXPLORATION CONSTANT

In addition to the parameters $\alpha$ and $\beta$ discussed in Section 4.2, the constant $c$ in DyRo-UCT, which balances exploration and exploitation, also needs to be tuned. We tested five settings $\{0, 1, 2, 3, 4\}$, and the results are shown in Figure 7. Based on the experimental results, setting the exploration constant $c = 3$ yields the best performance among the evaluated parameter configurations, achieving both the highest overall performance and lowest variance. We therefore set $c = 3$ in our experiment.

Among all the tested values, only $c = 0$ leads MCTS to perform worse than the offline-learned policies. In this case, the exploration term in UCT is entirely removed, resulting in a greedy search that focuses on areas previously yielding good results without sufficiently exploring less tried actions, ultimately leading to inferior search outcomes.

### A.2 ACTION SELECTION CRITERIA

After performing MCTS, two commonly employed criteria for selecting the next action to execute are choosing the action with either the highest value or the highest visit count. We conducted comparative experiments using these selection criteria on both PUCT-MCTS and DyRo-MCTS, with results presented in Figure 8.

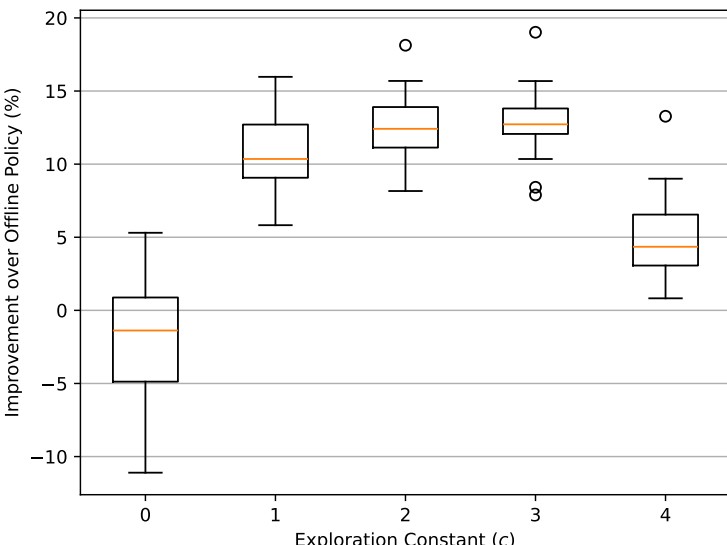

Figure 7: Performance improvement (the higher the better) of DyRo-MCTS under different settings of the exploration constant $c$.

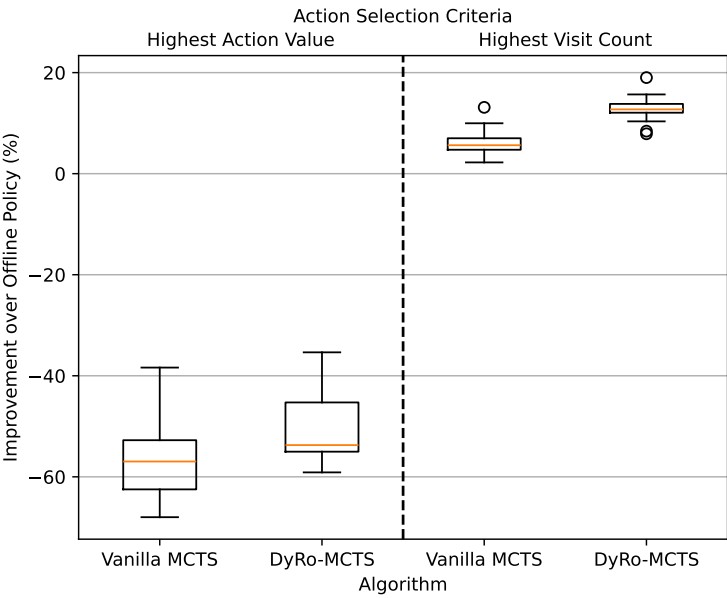

Figure 8: Performance comparison of PUCT-MCTS and DyRo-MCTS using highest visit count versus highest value as action selection criteria.

Experimental results indicate that selecting actions based on the highest visit count significantly outperforms selection based on the highest value in both PUCT-MCTS and DyRo-MCTS. It is likely because, in the lookahead planning for DJSS, high visit counts offer enhanced stability and confidence, representing actions that have been thoroughly explored and consistently associated with favourable outcomes during the search. In contrast, selecting based on action value can be unreliable due to noise from rare high-reward outcomes that skew the average. Therefore, we adopt the highest visit count as the action selection criterion in our experiments.

## B   RESULTS OF MANUALLY DESIGNED SCHEDULING HEURISTICS

In principle, any policy providing informed prior probabilities can guide the DyRo-MCTS algorithm. Manually designed scheduling heuristics are also widely used in production practice due to their ease of use and good interpretability. In this study, we collect ten well-known scheduling heuristics from prior literature and integrate them into our main experiments in Section 4.3. The specifications of these manually designed heuristics are presented in Table 3.

| Name | Description |
| --- | --- |
| SPT | Select the job with the shortest processing time for its current operation. |
| SWINQ | Select the job whose next operation will be executed on the machine with the lowest workload. |
| CR | Select the job with the highest Critical Ratio, calculated as total remaining processing time of a job divided by the time remaining until its due date. |
| SL | Select the job with the shortest slack time, calculated as the difference between the due date of a job and the current time. |
| ATC | Prioritise jobs based on their Apparent Tardiness Cost (Vepsalainen & Morton, 1987). |
| COVERT | Prioritise jobs based on their Cost Over Time (Carroll, 1965) |
| MOD | Prioritise jobs based on their Modified Operation Due date (Baker, 1974). |
| Anderson Rule | The CR + PT rule, proposed by Anderson & Nyirenda (1990). |
| Holthaus Rule 1 | The PT + WINQ + SL rule, proposed by Holthaus & Rajendran (2000) |
| Holthaus Rule 2 | The 2PT + WINQ + NPT rule, proposed by Holthaus & Rajendran (2000) |

Table 3: Ten manually designed scheduling heuristics adopted in Section 4.3.

In Section 4.3, we present the average scheduling results for the ten manually designed heuristics. In this section, detailed results for each heuristic are provided in Table 4. The results indicate that the SPT heuristic is generally the most effective for guiding MCTS, while Holthaus Rule 2 demonstrates superior performance when applied directly in scenarios $\langle T_{mean}, 0.85 \rangle$ and $\langle T_{mean}, 0.95 \rangle$.

## C   IMPACT OF OFFLINE POLICY QUALITY ON ONLINE PLANNING

To investigate the relationship between the scheduling performance of an offline policy and the DyRo-MCTS guided by the policy, an experiment was conducted using 100 offline-learned policies with varying performance. The results are presented in Figure 9. In the scatter plot, each point corresponds to a policy. The x-axis represents its scheduling performance (mean tardiness) when applied directly without online planning. The y-axis shows the performance when DyRo-MCTS is applied using the same policy as guidance.

The results indicate a clear trend: policies that exhibit stronger performance when used directly tend to yield better outcomes when used to guide DyRo-MCTS. This highlights the importance of learning high-quality policies offline, as they enable more informed and effective online planning.

## D   THE USE OF LARGE LANGUAGE MODELS

In this work, large language models are used solely for polishing the paper's writing.

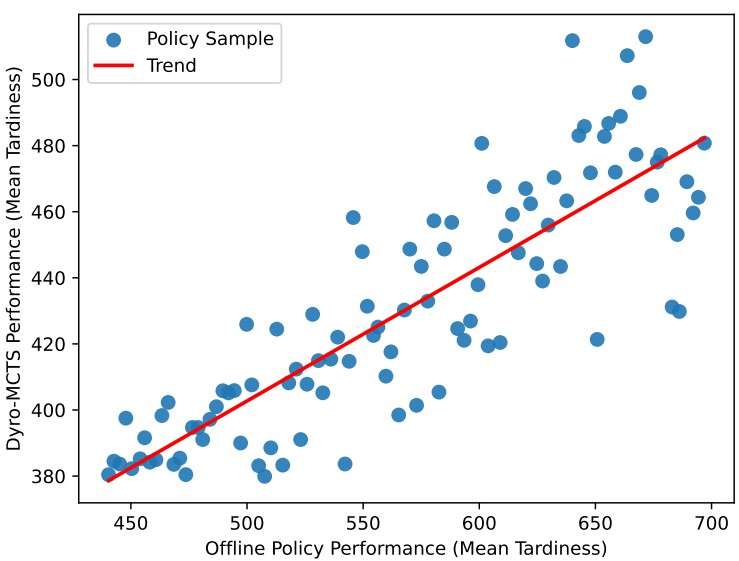

Figure 9: Correlation between offline policy performance and DyRo-MCTS performance under the policy guidance.

Table 4: Performance comparison of pure offline policy, PUCT-MCTS, and DyRo-MCTS guided by ten manually designed scheduling heuristics across different scheduling scenarios. The best results are highlighted in bold font. Symbols (↑) and (↓) indicate that one method is significantly better or worse than the previous method, respectively, while (≈) denotes no significant difference.

| Scenario | Policy | w/o Online Planning Perf. | + PUCT-MCTS Perf. | Imp. | + DyRo-MCTS Perf. | Imp. |
|---|---|---|---|---|---|---|
| $T_{mean}$ 0.85 | SPT | 509.46±65.96 | **434.33±58.41** (↑) | 15% | **389.53±46.84** (↑) (↑) | 23% |
| | SWINQ | 622.07±75.29 | 478.3±58.16(↑) | 23% | 433.5±49.35(↑)(↑) | 30% |
| | CR | 885.33±125.73 | 526.11±71.61(↑) | 40% | 459.42±57.19(↑)(↑) | 48% |
| | SL | 835.33±106.93 | 538.91±74.16(↑) | 36% | 480.67±62.13(↑)(↑) | 42% |
| | ATC | 589.21±67.24 | 463.08±59.72(↑) | 21% | 420.9±50.39(↑)(↑) | 28% |
| | COVERT | 604.81±69.87 | 468.69±62.36(↑) | 23% | 425.06±51.94(↑)(↑) | 30% |
| | MOD | 617.23±67.32 | 463.32±57.29(↑) | 25% | 422.61±49.19(↑)(↑) | 32% |
| | Anderson Rule | 870.67±129.42 | 524.91±72.21(↑) | 40% | 456.87±56.53(↑)(↑) | 47% |
| | Holthaus Rule 1 | 712.96±91.36 | 514.66±70.63(↑) | 28% | 462.79±58.76(↑)(↑) | 35% |
| | Holthaus Rule 2 | **489.04±56.42** | 435.98±58.43(↑) | 11% | 392.25±46.96(↑)(↑) | 20% |
| $T_{mean}$ 0.95 | SPT | 1827.3±496.93 | **1529.1±361.87** (↑) | 16% | **1282.36±321.24** (↑) (↑) | 29% |
| | SWINQ | 2059.33±534.73 | 1686.02±402.25(↑) | 18% | 1418.8±354.93(↑)(↑) | 31% |
| | CR | 3227.01±725.11 | 1915.6±431.15(↑) | 41% | 1514.28±356.19(↑)(↑) | 53% |
| | SL | 2693.57±556.18 | 1986.52±465.52(↑) | 26% | 1652.8±409.85(↑)(↑) | 39% |
| | ATC | 1906.19±500.54 | 1637.27±386.82(↑) | 14% | 1367.72±336.72(↑)(↑) | 28% |
| | COVERT | 1925.91±476.85 | 1648.42±382.99(↑) | 14% | 1378.23±340.58(↑)(↑) | 28% |
| | MOD | 1930.15±482.25 | 1557.79±348.95(↑) | 18% | 1327.61±318.73(↑)(↑) | 31% |
| | Anderson Rule | 3209.66±710.03 | 1924.32±444.01(↑) | 40% | 1511.91±362.7(↑)(↑) | 53% |
| | Holthaus Rule 1 | 2328.79±512.81 | 1884.88±440.02(↑) | 19% | 1595.34±405.07(↑)(↑) | 32% |
| | Holthaus Rule 2 | **1573.57±379.12** | 1570.43±377.01 (≈) | 0% | 1306.93±332.89(↑)(↑) | 17% |
| $WT_{mean}$ 0.85 | SPT | **811.39±91.8** | **784.22±95.86** (↑) | 3% | **723.36±84.2** (↑) (↑) | 11% |
| | SWINQ | 1366.82±165.94 | 964.44±118.62(↑) | 29% | 880.2±102.37(↑)(↑) | 36% |
| | CR | 1965.44±274.14 | 1097.37±147.67(↑) | 44% | 954.17±112.93(↑)(↑) | 51% |
| | SL | 1607.91±205.52 | 1055.76±145.83(↑) | 34% | 940.32±119.15(↑)(↑) | 41% |
| | ATC | 1043.77±103.25 | 832.16±102.24(↑) | 20% | 781.16±88.88(↑)(↑) | 25% |
| | COVERT | 1078.6±106.68 | 842.89±100.81(↑) | 22% | 790.15±90.61(↑)(↑) | 27% |
| | MOD | 1359.43±151.49 | 944.7±119.17(↑) | 30% | 855.68±99.14(↑)(↑) | 37% |
| | Anderson Rule | 1936.56±283.38 | 1084.39±144.21(↑) | 44% | 945.12±112.15(↑)(↑) | 51% |
| | Holthaus Rule 1 | 1568.72±203.53 | 1092.41±155.81(↑) | 30% | 973.55±127.41(↑)(↑) | 38% |
| | Holthaus Rule 2 | 1076.59±127.23 | 868.39±112.12(↑) | 19% | 783.36±92.63(↑)(↑) | 27% |
| $WT_{mean}$ 0.85 | SPT | **2459.32±572.74** | 2534.83±568.64 (↓) | -3% | **2212.25±527.42** (↑) (↑) | 10% |
| | SWINQ | 4534.57±1189.48 | 3364.79±817.01(↑) | 25% | 2806.71±693.61(↑)(↑) | 38% |
| | CR | 7093.34±1580.98 | 3699.36±734.31(↑) | 48% | 2935.71±624.14(↑)(↑) | 58% |
| | SL | 5036.34±1019.91 | 3732.71±839.43(↑) | 26% | 3137.71±753.14(↑)(↑) | 38% |
| | ATC | 2735.81±572.97 | 2579.33±546.03(↑) | 6% | 2292.64±500.94(↑)(↑) | 16% |
| | COVERT | 2795.83±567.64 | 2630.37±572.3(↑) | 6% | 2331.02±515.97(↑)(↑) | 17% |
| | MOD | 4234.3±1049.66 | 3009.71±650.16(↑) | 28% | 2578.67±599.25(↑)(↑) | 39% |
| | Anderson Rule | 7060.07±1597.3 | 3704.82±754.46(↑) | 47% | 2920.18±618.04(↑)(↑) | 58% |
| | Holthaus Rule 1 | 5125.93±1139.43 | 4008.16±921.02(↑) | 22% | 3362.17±838.67(↑)(↑) | 34% |
| | Holthaus Rule 2 | 3451.61±856.48 | 3081.29±742.67(↑) | 10% | 2554.37±637.16(↑)(↑) | 26% |