# OpenReview forum: "DyRo-MCTS: A Robust Monte Carlo Tree Search Approach to Dynamic Job Shop Scheduling"
_ICLR.cc/2026/Conference — Submitted to ICLR 2026_

### Official Review · Reviewer_hkab · 2025-10-30

**Soundness:** 2
**Presentation:** 2
**Contribution:** 1
**Rating:** 2
**Confidence:** 5

**Summary:**

The paper proposes DyRo-MCTS, a robust Monte Carlo Tree Search framework for dynamic job shop scheduling (DJSS). It aims to improve online scheduling decisions under uncertain job arrivals by incorporating an action robustness estimation term into the MCTS tree policy. The method estimates robustness through machine idle-time distributions and introduces a DyRo-UCT formula balancing exploitation, exploration, and robustness. Experiments show that DyRo-MCTS improves upon both offline-learned scheduling policies (from DRL and GP) and vanilla MCTS with minimal added computation.

**Strengths:**

The paper addresses a practical and important problem—robust online planning in dynamic scheduling—by extending MCTS with a simple, interpretable robustness measure. The formulation is clear and methodologically consistent.

**Weaknesses:**

The novelty of DyRo-MCTS is limited. Integrating a robustness term into MCTS, while intuitively reasonable, constitutes an incremental extension of existing robust scheduling and MCTS formulations. Prior reinforcement learning and online planning studies have already aimed to handle incomplete or uncertain job information, yet related work comparisons are not sufficiently comprehensive. The experiments focus mainly on comparing against vanilla MCTS and lack evaluation against other modern online planning or robust RL approaches. Claims about negligible computation overhead are not quantitatively analyzed beyond MCTS iteration scaling, and no ablation or complexity analysis supports this.

**Questions:**

The study claims to handle online planning under incomplete information, but why are comparisons with other online or robust RL-based planners absent?

The method states minimal computational cost increase—where is the quantitative evidence or runtime breakdown to substantiate this?

How does DyRo-MCTS differ conceptually from prior robust MCTS or uncertainty-aware planning approaches?

---

> ### Author Response · Authors · 2025-12-04
>
> We sincerely thank the reviewer for the valuable comments, which help us improve the quality of this work.
>
> **W1 & Q1: Prior reinforcement learning and online planning studies have already aimed to handle incomplete or uncertain job information, yet related work comparisons are not sufficiently comprehensive. The experiments focus mainly on comparing against vanilla MCTS and lack evaluation against other modern online planning or robust RL approaches. The study claims to handle online planning under incomplete information, but why are comparisons with other online or robust RL-based planners absent?**
>
> We compare the proposed DyRo-MCTS with two robust scheduling methods (one DRL-based and one GP-based) in Section 4.4. These two methods are designed to recognise the random job-arrival patterns in dynamic scheduling and to make robust decisions under incomplete information. In the revised manuscript, we have added two additional state-of-the-art methods, PSAO, and Li-MCTS, to make a more comprehensive comparison. The results show that DyRo-MCTS significantly outperforms above online and offline dynamic scheduling methods across all four scheduling scenarios.
>
> **W2 & Q2: Claims about negligible computation overhead are not quantitatively analysed beyond MCTS iteration scaling, and no ablation or complexity analysis supports this. The method states minimal computational cost increase—where is the quantitative evidence or runtime breakdown to substantiate this?**
>
> We have provided more details about the computation overhead in Section 4.4. As the table showing below. Compare with pure inference models DRL and GP, decision time of MCTS-based methods increase notably spent at the online planning process. Compared with PUCT-MCTS, DyRo-MCTS only increase 2 ms per decision for robustness estimation. However, compared with traditional online planner PSAO, the computation overhead of above mentioned methods decrease in magnitude.
>
> | **Algorithm**           | PSAO          | Li-MCTS | DRL   | GP                 | PUCT-MCTS | DyRo-MCTS |
> |-------------------------|---------------|---------|-------|---------------------|-----------|-----------|
> | **Decision Time (ms)** | $8.50\times10^3$ | 68.01   | 0.50  | $7.97\times10^{-3}$ | 20.53     | 22.30     |
>
> How much decision time can be considered as negligible is depends on the production case [1]. For example, in semiconductor scheduling scenario, decision within 1000 ms is considered as real-time [2]. In this standard, the decision of DyRo-MCTS can be considered as real-time.
>
> **W3 & Q3: The novelty of DyRo-MCTS is limited. Integrating a robustness term into MCTS, while intuitively reasonable, constitutes an incremental extension of existing robust scheduling and MCTS formulations. How does DyRo-MCTS differ conceptually from prior robust MCTS or uncertainty-aware planning approaches?**
>
> Make robust decision in DJSS is non-trivia for MCTS methods and is not properly addressed in existing studies. Explicitly modelling this transition uncertainty using existing methods [3] is challenging due to the infinite possibilities of new job arrivals. Sample-based approaches [4] are also impractical due to the NP-hard nature of DJSS. Each randomly introduced job factorially expands the search space, hindering real-time decision-making. Therefore, existing MCTS-based dynamic scheduling methods often compromise by restricting their online planning to existing jobs.
>
> ---
>
> [1] D. Ouelhadj and S. Petrovic, ‘A survey of dynamic scheduling in manufacturing systems’, J Sched, vol. 12, no. 4, pp. 417–431, Aug. 2009.
>
> [2] A. T. H. Ang, ‘On-line near-real-time scheduling of semiconductor wafer fabrication’, Nanyang Technological University, 2008.
>
> [3] F. Kohankhaki, K. Aghakasiri, H. Zhang, T.-H. Wei, C. Gao, and M. Müller, ‘Monte Carlo Tree Search in the Presence of Transition Uncertainty’, AAAI, vol. 38, no. 18, pp. 20151–20158, Mar. 2024.
>
> [4] D. Silver and J. Veness, ‘Monte-Carlo Planning in Large POMDPs’, Advances in neural information processing systems, vol. 23, 2010.

---

### Official Review · Reviewer_F8EK · 2025-10-31

**Soundness:** 2
**Presentation:** 2
**Contribution:** 2
**Rating:** 4
**Confidence:** 4

**Summary:**

The paper augments MCTS for dynamic JSS (with new job arrivals) by adding a robustness score based on integrated idleness to better control weighted tardiness. An offline policy supplies action priors, and partial tree reuse is used to keep per-decision latency low.

**Strengths:**

The robustness term captures a practical scheduling intuition—discouraging early idle time—that is relevant when arrivals are uncertain and the objective is (weighted) tardiness.

The method needs minimal additional online planning time.

**Weaknesses:**

The robustness score is heuristic and domain-dependent; there is no theoretical link to optimality or selection consistency.

Coverage of scenario diversity is limited; the paper lacks analysis across varied WIP levels and machine counts.

The description of tree reuse in a dynamic environment (e.g., around line 248) is insufficient. It is unclear what statistics are retained/reset and whether reuse was actually enabled in experiments.

**Questions:**

1. Do you have a theoretical justification for reusing nodes when the state changes due to new arrivals? Is carrying over statistics from prior rollouts purely pragmatic? What exactly is retained (Q-values, visit counts, priors, normalization ranges), and what is reset?
2. For the first 1,000 arrivals, are the schedules identical across “w/o online planning”, “vanilla MCTS”, and “DyRo-MCTS”?
3. What is the typical number of jobs considered per decision during MCTS? Also, can you report results for settings with ≥100 concurrent jobs (WIP level) and ≥10 machines?
4. With 100 MCTS iterations and non-parallel rollouts, you report that MCTS takes around 0.021 seconds per decision. Are GP (Chen et al., 2025) and DRL (Liu et al., 2023) dispatchers in a similar decision time?

---

> ### Author Response · Authors · 2025-12-04
>
> We truly thank the reviewer suggestions in making our experiments more comprehensive and make our conclusions more convincible.
>
> **W1: The robustness score is heuristic and domain-dependent; there is no theoretical link to optimality or selection consistency.**
>
> We generally agree with the reviewer. As with most robustness measures in machine scheduling [1], this estimation is heuristic in nature and difficult to connect to any theoretical proof. Nonetheless, our extensive experiments demonstrate its empirical effectiveness and clear advantages.
>
> Although the robustness estimation is tailored to the dynamic scheduling domain rather than broader MCTS domains, it provides substantial value. DJSS is a fundamental requirement in modern manufacturing. We believe our work sheds meaningful light on this important domain.
>
> **W2: Coverage of scenario diversity is limited; the paper lacks analysis across varied WIP levels and machine counts.**
> **Q3: What is the typical number of jobs considered per decision during MCTS? Also, can you report results for settings with ≥100 concurrent jobs (WIP level) and ≥10 machines?**
>
> We added a new experiment in Section 4.4 of the revised manuscript covering four problem scales. The number of machines was set to 5, 10, 15, 20. The anticipated machine utilisation is fixed at 0.95, so the WIP level increases as the number of machines grows. We recorded the WIP at each decision point and obtained the following statistics.
>
> | # Machines | WIP |
> |----------|----------|
> | 5 | 43.4±25.0 |
> | 10 | 66.3±29.5 |
> | 15 | 104.5±45.5 |
> | 20 | 126.1±45.9 |
>
> The performance comparison with SOTA dynamic scheduling algorithms is presented in Figure 6 of the revised manuscript. The results show that, as the problem becomes more complex (with more machines and higher WIP), the mean job tardiness increases for all algorithms. However, DyRo-MCTS consistently achieves significantly better performance than its competitors.
>
> **W3: The description of tree reuse in a dynamic environment (e.g., around line 248) is insufficient. It is unclear what statistics are retained/reset and whether reuse was actually enabled in experiments.**
> **Q1: Do you have a theoretical justification for reusing nodes when the state changes due to new arrivals? Is carrying over statistics from prior rollouts purely pragmatic? What exactly is retained (Q-values, visit counts, priors, normalization ranges), and what is reset?**
>
> In Section 3.2, we have added a more detailed description of subtree keeping strategies:
>
> When no new job arrives between two consecutive decision points, the entire search tree from the subsequent state is retained, and all statistics, including total action value \(w(s,a)\), visit count\(n(s)\), prior \(p(s,a)\), and normalisation ranges, are preserved. Once a new job arrives, the whole search tree is discarded, as the previous rollouts do not contain the operations of the new job, so their statistics are not reusable.
>
> **Q2: For the first 1,000 arrivals, are the schedules identical across “w/o online planning”, “vanilla MCTS”, and “DyRo-MCTS”?**
>
> Yes, the schedules of the first 1,000 arrivals are identical. During the warm-up phase, jobs are dispatched using the SPT rule. All comparison methods start their schedules from the same initial state.
>
> **Q4: With 100 MCTS iterations and non-parallel rollouts, you report that MCTS takes around 0.021 seconds per decision. Are GP (Chen et al., 2025) and DRL (Liu et al., 2023) dispatchers in a similar decision time?**
>
> We have added a detailed analysis of decision time is provided in Section 4.4 of the revised manuscript. On average, the decision times of GP ($7.97 \times 10^{-6}$ seconds) and DRL ($5.03 \times 10^{-4}$ seconds) are much lower than those of the MCTS-based methods, as they make decisions through model inference rather than online search. However, such differences are negligible in practice, as they are all much less than a second.
>
> ---
>
> [1] N. Zhang, Y. Zhang, S. Song, and C. L. P. Chen, ‘A Review of Robust Machine Scheduling’, IEEE Trans. Automat. Sci. Eng., vol. 21, no. 2, pp. 1323–1334, Apr. 2024.

---

### Official Review · Reviewer_M7jB · 2025-11-01

**Soundness:** 3
**Presentation:** 3
**Contribution:** 2
**Rating:** 2
**Confidence:** 5

**Summary:**

The paper studies a dynamic job shop scheduling problem (with dynamically arriving jobs), and proposes a monte carlo tree search (MCTS) method for online planning. The proposed method DyRo-MCTS integrates action robustness estimation in the tree policy of MCTS. In the experiments, the authors show the DyRo-MCTS outperforms a vanilla MCTS and improves the performance of several offline learned policies.

**Strengths:**

1.	Promising performance compared to offline policies and vanilla MCTS

2.	Good analysis for parameter sensitivity

**Weaknesses:**

1.	The authors review some MCTS applications in scheduling and note that mentioned ones are designed for static JSS. The proposed method differs from those for the targeted dynamic JSS. However, there are existing work on MCTS for dynamic JSSP that the authors may overlook such as [1-6]:

[1] Li, Kexin, et al. "An effective MCTS-based algorithm for minimizing makespan in dynamic flexible job shop scheduling problem." Computers & Industrial Engineering 155 (2021): 107211.

[2] Saqlain, M., S. Ali, and J. Y. Lee. "A Monte-Carlo tree search algorithm for the flexible job-shop scheduling in manufacturing systems." Flexible Services and Manufacturing Journal 35.2 (2023): 548-571.

[3] He, Zhou, et al. "Enhanced Monte‐Carlo tree search for dynamic flexible job shop scheduling with transportation time constraints." Expert Systems 42.2 (2025): e13727.

[4] Kim, Duyeon, and Hyun-Jung Kim. "Monte Carlo tree search-based algorithm for dynamic job shop scheduling with automated guided vehicles." 2022 Winter Simulation Conference (WSC). IEEE, 2022.

[5] Cheng, Yuxia, et al. "Smart DAG tasks scheduling between trusted and untrusted entities using the MCTS method." Sustainability 11.7 (2019): 1826.

[6] Li, Weiguan, Jialun Li, and Weigang Wu. "A dynamic scheduling algorithm with time varying resource constraints in colocation data centers." 2022 13th International Conference on Information and Communication Systems (ICICS). IEEE, 2022.

2. The key contribution of the work is to re-define the exploitation term with the weighted sum of action value and robustness, which brings a limited contribution, given that applying MCTS for dynamic scheduling is not new.

3. In addition, the weighted term looks similar to RAVE [7] and other variants of UCT formula:

[7] Gelly, Sylvain, and David Silver. "Combining online and offline knowledge in UCT." Proceedings of the 24th international conference on Machine learning. 2007.

4. Experiments only compare with offline policies.  Including classic online methods for dynamic scheduling are needed. It is not clear for the DRL baseline, whether the greedy or sampling strategy is used during inference.

**Questions:**

1.	Why does DyRo-MCTS allow jobs to be delayed, whereas MCTS does not?

2.	What is the setting (UCT terms?) of vanilla MCTS?

---

> ### Author Response · Authors · 2025-12-04
>
> We sincerely thank the reviewer for acknowledging the soundness and presentation of this paper.
>
> **W1: Several MCTS studies for DJSS [1-6] already exist.**
>
> **W2: The key contribution—redefining the exploitation term as a weighted sum of action value and robustness—offers limited novelty, as applying MCTS to dynamic scheduling is not new.**
>
> While several existing works have applied MCTS to DJSS, the focus of this study is fundamentally different. Prior research largely falls into two categories:
> (1) adopting established MCTS techniques (such as RAVE and multi-branching simulation) to accelerate search and improve efficiency, as seen in [1, 3] from the reviewer's comment; and
> (2) adapting MCTS to specific DJSS variants, including flexible machine settings [2], transportation-time constraints [3], automated guided vehicles [4], and related task-scheduling problems [5, 6].
>
> In contrast, this work aims to enable MCTS with robust decision-making capability under continuous disturbances. A longstanding limitation in existing studies is that, at each decision point, they simply solve a static scheduling problem. However, relying solely on existing job information for planning often produces schedules that lack the flexibility required to accommodate future job arrivals (as illustrated in Section 4.6 of the revised manuscript). Addressing this issue is non-trivial. The transition uncertainty in DJSS is very difficult to model explicitly through existing uncertainty-adapted MCTS variants [1], due to the infinite possibilities of new job arrivals.
>
> To address this challenge, we introduce a robustness estimation term into the PUCT formula, designed specifically for dynamic scheduling. This modification is easy to implement, and we regard its simplicity as an advantage. Extensive experiments show that this adjustment effectively overcomes the key limitation of SOTA MCTS approaches and yields substantial performance improvements. Its simplicity also makes it easy to combine with other MCTS enhancements proposed in earlier work.
>
> **W3: The weighted term looks similar to RAVE and other variants of the UCT formula.**
>
> The exploration term $ \mathcal{E}(s,a) = \alpha, q(s,a) + (1-\alpha), \rho(s,a) $ in DyRo-UCT indeed looks similar to RAVE in form. It uses linear interpolation to balance the empirical mean reward $ q(s,a) $ with the robustness estimate $ \rho(s,a) $. Although linear interpolation is widely used in many algorithms beyond UCT, its purpose here is fundamentally different.
>
> For example, the RAVE method uses interpolation to shift from its fast but biased estimates $Q_{\mathrm{RAVE}}$ to the slower but unbiased UCT estimates $Q_{\mathrm{UCT}}$ as visit counts increase.
>
> **W4: Including classic online methods for dynamic scheduling is needed. It is also unclear whether the DRL baseline uses greedy or sampling during inference.**
>
> We have added a new experiment comparing DyRo-MCTS with a classic online method named PSAO [2], which combines particle swarm optimisation with arithmetic optimisation to improve both schedule quality and robustness. We also included another state-of-the-art MCTS-based method beyond PUCT, namely Li-MCTS [3]. The results in Figure~6 of the revised manuscript show that DyRo-MCTS consistently outperforms these classic online methods across four different scheduling scenarios.
>
> A greedy strategy is used for DRL policy inference, following the original design of the adopted DRL method.

---

> > ### Author Response · Authors · 2025-12-04
> >
> > **Q1: Why does DyRo-MCTS allow jobs to be delayed, whereas MCTS does not?**
> >
> > In fact, all the compared MCTS methods allow jobs to be delayed in our experiment, and the comparison is fair. The difference is that, compared with other MCTS algorithms, DyRo-MCTS is more tolerant of job delays in order to maintain a robust schedule. In dynamic scheduling, a schedule is unlikely to be executed exactly as planned; when new jobs arrive and must be incorporated, the overall tardiness of a robust schedule does not change significantly. In contrast, a non-robust schedule may initially appear to delay fewer jobs, but its quality can deteriorate sharply once new jobs need to be integrated.
> >
> > **Q2: What is the setting (UCT) of the vanilla MCTS?**
> >
> > The vanilla neural MCTS algorithm adopts PUCT: $a^* = \arg\max_a \left[, q(s,a) + c \cdot p(s,a), \frac{\sqrt{n(s)}}{1+n(s,a)} \right],$
> > where the exploration constant ($c$) is set to 3 according to parameter tuning. The prior (p(s,a)) is obtained from offline-learned DRL or GP policies.
> >
> > ---
> >
> > **References**
> >
> > [1] F. Kohankhaki et al., *Monte Carlo Tree Search in the Presence of Transition Uncertainty*, AAAI, 38:20151–20158.
> >
> > [2] J. Duan and J. Wang, *Robust scheduling for flexible machining job shop subject to machine breakdowns and new job arrivals considering system reusability and task recurrence*, Expert Systems with Applications, 203:117489, 2022.
> >
> > [3] K. Li et al., *An effective MCTS-based algorithm for minimising makespan in dynamic flexible job shop scheduling problem*, Computers & Industrial Engineering, 155:107211, 2021.

---

### Official Review · Reviewer_YbgJ · 2025-11-01

**Soundness:** 4
**Presentation:** 4
**Contribution:** 4
**Rating:** 8
**Confidence:** 4

**Summary:**

The paper proposes to modify MCTS for the dynamic JSSP so as to improve robustness of the decisions. The robustness is defined as the early idleness of jobs and MCTS combines the usual Q value with robustness. Experiments show that the approach is beneficial and performs better than MCTS with different playout policies.

**Strengths:**

A novel MCTS algorithm for dynamic JSSP
The contribution is clear
Good experimental results

**Weaknesses:**

No comparison to other Monte Carlo Search algorithms than PUCT.

**Questions:**

When combining with GP do you use the GP policy as the prior in the PUCT formula?

---

> ### Author Response · Authors · 2025-12-04
> **Expanded Experimental Comparisons with SOTA MCTS Methods**
>
> We sincerely appreciate the reviewer's strong acknowledgement of the quality and contribution of this work. We also find the suggestions highly valuable for further improving the study. Accordingly, we have provided a more comprehensive comparison with other state-of-the-art MCTS algorithms.
>
> **Weakness: No comparison to other Monte Carlo Search algorithms beyond PUCT.**
>
> We have included an additional state-of-the-art MCTS method beyond PUCT for DJSS, i.e., Li-MCTS [1], into comparison. Li-MCTS incorporates several advanced techniques, such as RAVE and rolling time window, along with a carefully designed subtree-keeping mechanism and an effective strategy for initialising and updating prior knowledge. DyRo-MCTS has now been compared with five state-of-the-art robust scheduling methods (as shown in Sections 4.4). DyRo-MCTS achieves significantly better performance across a wide range of scheduling scenarios. This improvement verifies the contribution of DyRo-MCTS in combining robust action-estimation mechanism with the search prior provided by state-of-the-art robust scheduling policies [2,3].
>
> **Question: When combining with GP, is the GP policy used as the prior in the PUCT formula?**
>
> Yes. The GP policy is used as the prior in the PUCT formula. GP learns symbolic functions that map state features to a priority score for each action.
>
> ---
>
> **References**
>
> [1] K. Li, Q. Deng, L. Zhang, Q. Fan, G. Gong, and S. Ding. *An effective MCTS-based algorithm for minimizing makespan in dynamic flexible job shop scheduling problem*. Computers & Industrial Engineering, 155:107211, 2021.
>
> [2] R. Liu, R. Piplani, and C. Toro. *A deep multi-agent reinforcement learning approach to solve dynamic job shop scheduling problem*. Computers & Operations Research, 159:106294, 2023.
>
> [3] R. Chen, Y. Mei, F. Zhang, and M. Zhang. *Neural network surrogate based on binary classification for assisting genetic programming in searching scheduling heuristic*. PRICAI 2024, LNCS 15281, pp. 309–321, 2025.

---

### Author Response · Authors · 2025-12-04
**General Response**

We sincerely appreciate the reviewer’s valuable comments, which have helped us further improve this work. We are also grateful for the reviewer’s acknowledgement of the soundness of our method design, the quality of our experimental study, the strong performance achieved by our approach, and the clarity of our presentation.

At the same time, we fully understand the concern regarding the contribution of this work relative to existing studies. We therefore wish to clarify and highlight the value offered by our approach.

This work identifies a fundamental yet largely overlooked issue in existing MCTS methods for DJSS: current MCTS approaches plan only on the jobs that exist in the environment. As a result, they fail to maintain schedules robust to disturbances from new job arrivals. This limitation persists because the transition uncertainty in DJSS is effectively intractable for existing uncertainty-aware MCTS variants, given the unbounded possibilities of new job arrivals.

Our contribution lies in addressing this gap with a simple, effective, and easy-to-implement robustness estimation method tailored for dynamic scheduling. Despite its simplicity, this modification directly resolves the core shortcoming of current MCTS approaches and delivers clear and consistent performance improvements across diverse dynamic scenarios. We believe this demonstrates both the practicality and the significance of our contribution to robust planning of MCTS in DJSS.

---

### Meta-Review · Area_Chair_RBp2 · 2026-01-11

**Summary:**

The paper proposed DyRo-MCTS for the dynamic JSSP (DJSSP). Overall, the reviewers recognize the strong experimental performance and the relevance of the problem of robust planning for the scheduling problems.

In a comprehensive rebuttal, the authors addressed several concerns by providing some additional experiments and elaborating the discussion on related works. However, several of the reviewers’ main concerns prevail.

The main concerns focus on the limited novelty of the work (only a robustness term is added to the UCT) and the limited comparison with other works that use MCTS approaches to the DJSSP. Although the authors discussed these and also added additional baseline approaches, the concerns remain. The applicability of the DyRo-UCT to a broad range of scenarios is insufficiently motivated to address a broad impact for the field. On the other hand, the experimental comparisons with other MCTS approaches for the DJSSP are not sufficient to establish major state-of-the-art results for the DJSSP problem. Hence, the main contribution to either the broader AI field or the more specific DJSSP field are not sufficiently motivated for inclusion in ICLR.
Hence, I propose to reject this paper and encourage the authors to further improve their paper.

**Reviewer Concerns:**

Several reviewer concerns have been addressed, such as the limited comparison to another MCTS algorithm beyond PUCT (YbgJ), limited discussion of other MCTS for dynamic JSSP works (M7jB), only comparing with offline baseline policies (M7jB), the description of tree reuse (F8EK), and unknown computational overhead (hkab).
However, some concerns remain such as the limited novelty (M7jB, hkab), lacking comparison with other MCTS approaches to the DJSSP (M7jB, hkab), and the robustness score being too domain-dependent (F8EK).

**Reviewer Scores:**

I believe reviewer YbgJ would keep its score of an 8. Reviewer F8EK might have changed their score to a 6, but may also have kept their score of 4. Reviewer M7jB and hkab might have raised their score to a 4, but significant concerns remain for both.

---

### Decision · Program_Chairs · 2026-01-26

Reject